# Sex Differences in Immune Responses to Infectious Diseases: The Role of Genetics, Hormones, and Aging

**DOI:** 10.3390/diseases13060179

**Published:** 2025-06-07

**Authors:** Pierluigi Rio, Mario Caldarelli, Edoardo Miccoli, Giulia Guazzarotti, Antonio Gasbarrini, Giovanni Gambassi, Rossella Cianci

**Affiliations:** 1Department of Medical and Surgical Sciences, Catholic University of Sacred Heart, 00168 Rome, Italy; pierluigi.rio01@icatt.it (P.R.); mario.caldarelli01@icatt.it (M.C.); edoardo.miccoli01@gmail.com (E.M.); giulia.guazzarotti1@gmail.com (G.G.); antonio.gasbarrini@unicatt.it (A.G.); giovanni.gambassi@unicatt.it (G.G.); 2Fondazione Policlinico Universitario A. Gemelli, Istituto di Ricerca e Cura a Carattere Scientifico (IRCCS), 00168 Rome, Italy

**Keywords:** gender medicine, genetics, hormones, immune system, infectious diseases

## Abstract

In recent years, gender medicine has emerged as a field of research analyzing sex-related differences in health and disease. Biological sex, depending on sex chromosome complement, sex steroid hormones, and reproductive organs, has been demonstrated to influence human susceptibility to infections, immune responses against pathogens, the clinical severity of infectious diseases, and responses to the available treatments. Men and women differ in their chromosome set, with men having one X chromosome (XY) and women two (XX). This different genetic composition results in a sex-dimorphic expression of genes and pathways involved in immune regulation, as well as in shaping immune responses to infectious agents. Moreover, estrogen, progesterone, and testosterone, impacting cells and pathways involved in both innate and adaptive immunity, have been shown to drive sex dimorphism in infectious diseases. This narrative review aims to explore the sex-related differences in responses to infections, specifically focusing on the underlying genetic and hormonal mechanisms. Hence, aging-related changes in the immune system and their potential impact on immune responses against pathogens will be discussed. Understanding sex differences and stratifying the population according to them will open the door to precision medicine and personalized patient care.

## 1. Introduction

The design of infectious disease transmission models is a crucial public health concept for understanding disease dynamics, outbreak prediction, and intervention analysis. These models connect important biological and epidemiological processes to generate quantitative estimates of transmission rates, reproduction numbers, and herd immunity thresholds and therefore projections of outbreaks and assessments of intervention strategies [1]. Traditionally, modelers make homogeneous population assumptions that can miss important key real-world heterogeneity in transmission or susceptibility or disease severity across population stratifications. Demographic factors, including age, sex, and socioeconomic status, influence contact patterns and exposure risk; in addition, these variables can greatly improve the accuracy of models and the design of more effective public health interventions [2].

While an increasing number of studies are highlighting sex differences in transmission dynamics, susceptibility, disease severity, and mortality outcomes, these dimensions remain under-represented in many infectious disease models [3]. Integrating these elements means pulling apart biological sex, based on genetic, cellular, and physiological traits, from gender, which acts as a social determinant of health. These two dimensions interplay as interactive and co-causal pathways that jointly determine disease risk, trajectory, treatment response, and outcomes.

Infectious disease outcomes are determined by biological sex-specific factors, including differences in viral susceptibility, immune responses, disease progression, and responses to anti-infective or anti-inflammatory therapies [4]. At the same time, gender operates as a set of social roles, expectations, and norms that create differential access to resources, opportunities, and risk exposures. These gendered dynamics play a major role in health inequities [5]. The interaction between genetics, hormonal, and social factors in modulating infections is reported in Figure 1.

We review the major sex differences in responses to infectious disease and to what extent such differences can be attributed to genetics due to the different sex chromosomes carried, hormones, and the immune response driven by each of these factors.

We performed a literature review of journal articles published in English in the last 15 years. The searches were conducted on PubMed, MEDLINE, and the Cochrane databases using the following medical subject headings: “Sex differences”, “Sex-mediated immunity”, “Sex hormones and immunity”, and “Sex-dimorphism in infectious diseases”. We also manually screened the references of the selected articles. The following were excluded: book chapters, conference proceedings, case reports, studies of the pediatric population, and those that were not available in full text.

## 2. Sex Differences in the Immune Response to Infectious Diseases: The Perspective of Genetics

Biological sex affects the functioning of all physiological systems, and the immune system is no exception. Men and women also differ in immune cell activity and self and foreign antigen response, and these variations are implicated in disease development and progression and immune responses. Genetically, men and women are different, with men having one X and one Y chromosome (XY) and women having two X chromosomes (XX).

Although one of the smallest human genome chromosomes and relatively gene-poor in terms of coding for proteins [6], the Y chromosome does take an active part in governing genes by restructuring chromatin through variation in its multicopy ribosomal gene sequences [7]. Myllymäki et al. convincingly demonstrated that the immune deficiency (IMD) pathway, a highly conserved NF-κB signaling cascade operating in *Drosophila melanogaster*, is subject to Y-linked regulatory variation (YRV). Through the introgression of distinct Y chromosomes into a common isogenic background, the authors generated male lines differing only in their Y chromosome. Such Y-line males displayed remarkable differences in immune gene expression and survival against *Serratia marcescens* infection, suggesting that polymorphisms on the Y chromosome can affect transcriptional regulation of immune genes and thereby contribute to sexually dimorphic immune responses [8]. Case et al. showed that genetic variations in the Y chromosome influence the susceptibility of B6 Y in mice to experimental autoimmune encephalomyelitis, with clear dimorphism [9]. Recently, hematopoietic mosaic loss of the Y chromosome (mLOY) has been identified as a potential male-specific factor that accelerates biological aging and increases the risk of several diseases [10]. mLOY was found to increase exponentially with age in circulating leukocytes [11]. Recent research has uncovered an association between hematopoietic mLOY and an increased risk of several age-related diseases, including Alzheimer’s and cardiovascular disease, diabetes mellitus, macular degeneration, and non-hematologic cancers [12]. This suggests that mLOY may act as a “causal” risk factor for these age-related diseases [13].

The expression of CD99, encoded on the Y chromosome in pseudo-autosomal region 1 (PAR1), was significantly reduced at both the transcript and protein levels in various immune cells [14]. The reduction in LOY may interfere with normal immune cell migration because CD99 is essential for transendothelial migration of immune cells [14]. Furthermore, LOY has been associated with the overexpression of nearly 500 autosomal genes in leukocytes, validating the hypothesis that mLOY in blood cells may influence the severity of disease by impairing immune cell function [15]. Additionally, Dumanski et al. described an LOY-associated transcriptional effect (LATE) on approximately 500 autosomal genes, highlighting leukocyte gene expression dysregulation with LOY [15]. The proportion of LATE genes within specific cell types was significantly greater than the proportion of LATE genes shared among different subsets of leukocytes, suggesting that LOY may have pleiotropic effects.

The X chromosome contains nearly 50 genes involved in immune function. Several genes are important for immune cell identity (e.g., FOXP3), activation and signaling (e.g., CD40L, TLR7, IRAK1, IL13RA1/2, NEMO, TASL, IL-9R), leukocyte migration (e.g., CD99, CXCR3), differentiation and growth (e.g., IL-2RG, BTK), and metabolism (e.g., OGT, CYBB) [16]. The gene TLR7 on Xq22.3 of the X chromosome [17] causes stimulation to activate the Myddosome complex, which includes MyD88 and IRAKs, with IRAK4 activating IRAK1, and this in turn signals through TRAF6 [18,19]. Similarly, the IRAK1 gene localizes to Xq28 [20]. Upon activation, TRAF6 stimulates interferon regulatory factor 7 (IRF7), which drives the transcription of type I interferons, including interferon (IFN)-α and IFN-β [21]. The BTK gene at Xq21.3-q22 interacts with and phosphorylates IκB-α to facilitate activation of the NF-κB pathway, a critical signaling cascade involved in immune response regulation, inflammation, and cell survival [22]. The X-linked ectodermal dysplasia receptor (XEDAR) is a newly identified protein that belongs to the tumor necrosis factor α (TNF) receptor family and is capable of activating both the canonical and non-canonical NF-κB pathways, contributing to diverse cellular processes, such as immune response, inflammation, and development [23].

The major genes and pathways of the immune system under the control of the sex chromosomes are listed in Table 1.

In addition, sex bias is influenced not only by X-linked genes but also by X-linked regulatory mechanisms, such as non-coding microRNAs (miRNAs) [24]. In particular, the X chromosome encodes approximately 10% of all genomic miRNAs.

To maintain equal gene dosage between the sexes, one of the X chromosomes is inactivated in women [25]. This inactivation is regulated by epigenetic processes and is characterized by repressive epigenetic features, such as long non-coding RNAs (lncRNAs) along with increased levels of DNA methylation and histone modifications [26]. The choice of which X chromosome to inactivate is random, with Xist (X-inactive-specific transcript) RNA playing a key role in determining which chromosome to silence [19]. Of the roughly 54 immunity-related genes found on the X chromosomes of humans and mice, 20.4% exhibit higher expression in females, while 16.7% have higher expression in males [27]. However, these percentages are likely underestimates due to the methodological challenges in accurately measuring transcription from the inactive X chromosome (Xi). Furthermore, immune cells exhibit deviations from the typical epigenetic features associated with the maintenance of canonical X chromosome inactivation (XCI) [28]. Specifically, mouse splenic B cells and T cells, and lymphocytes circulating in human blood, lack detectable Xist RNA and a build-up of heterochromatic modifications on the Xi, even though no compelling alterations in Xist transcription have been reported [29]. The non-canonical XCI maintenance characterizes both innate and adaptive immune cells. The distribution of Xist RNA to the inactive X chromosome varies greatly with the immune cell type [29]. Neutrophils and plasmacytoid dendritic cells (pDCs) do not show detectable Xist RNA transcripts at the Xi. In contrast, natural killer (NK) cells show clusters of Xist RNA within the Xi region [29].

Sex differences can be found in the incidence, symptoms, progression, and treatment outcome of many diseases including infections, autoimmunity, cardiovascular diseases, and cancer. These differences are associated directly with differences in immune responses, which develop and change throughout life in normal physiological states and pathology. Genes on the X and Y chromosomes may underlie sex-based differences in disease susceptibility. These immune-relevant genes, like TLR7, IRAK1, and immune-related miRNAs, are often found on the X chromosome [30]. Although X inactivation balances gene dosage in women, some regions evade inactivation, causing transcriptional upregulation of immune-related genes that contribute to sex-specific responses [30]. The Y chromosome has a function in the control of immune gene expression and infection susceptibility [31]. It influences the body’s susceptibility to Coxsackie virus infection regardless of testosterone levels and the immune response to influenza A virus infection [32].

Some sex-dependent loci co-localize, indicative of clusters of functionally related genes or genes involved in regulating multiple infections [33]. For example, Tmevd7 on chromosome 5 is located in an area associated with Chlamydia susceptibility, while Cnes2 on chromosome 17 is located in an area associated with both Chlamydia and influenza susceptibility. Furthermore, Cnes3 on the end of chromosome 17 overlaps with Lmr27, also lending credence to the existence of shared genetic factors that influence immunity [34]. Other loci, including Lmr15 and Tmevd6, show sex-specific effects in a “flip-flop” way: the same locus has opposing directions of effects in men and women [35]. These loci cover long chromosomal segments, so it is possible that there could be two tightly linked genes, one exhibiting effects in men and one in women or a single gene with antagonistic effects. A similar phenomenon occurs in humans: the IL-9 polymorphism rs2069885 (c.350 C>T) shows different effects on risk for severe respiratory syncytial virus infection in boys and girls [36]. Moreover, IL-9 variations impact *Aspergillus fumigatus*-mediated allergic lung inflammation via sex-dependent mechanisms, within the context of IL-9/IL-9R interactions [37]. IL-9 and its receptor IL-9R located on region 2 (PAR2) on Xq28 and Yq12 may contribute to sex-based differences. Given that PAR regions escape X inactivation, and recombine during meiosis, they act as autosomal genes and, therefore, may affect sex-specific immune responses [38].

Regarding viruses, data have indicated that men have higher COVID-19 mortality rates and worse outcomes [39]. Notably, the ACE2 gene that encodes an essential receptor for SARS-CoV-2 entry into cells is located on the X chromosome (Xp22.2) [40]. Women, endowed with two gene copies, can be heterozygous, whereas men are hemizygous. This genetic difference, coupled with X chromosome mosaicism, has been suggested as a factor in women’s improved capacity to fight SARS-CoV-2 infection [41].

Chromosomal polymorphisms influence the severity of illness. For instance, a single-nucleotide polymorphism on the X chromosome is linked to slower AIDS progression in women but not in men [42]. Furthermore, polymorphisms in the TLR8 gene on the X chromosome, which is associated with tuberculosis susceptibility, is linked to boys [43].

In addition to sex chromosomes, genetic variation also contributes to sexual dimorphism in immune responses. For example, polymorphisms in the IL-6 promoter are associated with chronic hepatitis C virus (HCV) infection in the mainly male cohort, whereas polymorphisms of the CTLA-4 gene associated with the clearance of HCV are seen more commonly in women [44].

## 3. Sex-Related Hormonal Differences in Immune Responses to Infectious Diseases

Among the factors determining biological sex differences, the concentration of sex hormones plays a pivotal role. Sex steroid receptors are largely distributed across different cells and tissues, thus influencing all physiological functions, including the immune one [45]. Hormone-related sex dimorphism has been identified in both innate and adaptive immunity, with estrogen usually enhancing the immune responses, and testosterone exerting an immunosuppressive effect [46,47]. Sex hormones have been proven to be drivers of the sex dimorphism in infectious and autoimmune diseases, with women generally being more susceptible to autoimmune diseases and less vulnerable to infectious diseases than men [48], which possibly implies a lower disease burden in women. For instance, the sex disparity in immune response to infections has been recently shown by the SARS-CoV-2 pandemic, which pointed out that higher levels of estrogen are associated with better clinical outcomes [49]. In the next paragraphs, the main effects of sex hormones on immune cells and pathways involved in the host response to pathogens will be described.

### 3.1. Estrogen

In the last few decades, estrogen has been largely studied for its impact on the immune system. The circulating estrogen levels differ between men and women during their lifetimes, with women typically having higher concentrations of these hormones during their reproductive years. Apart from its effects on cellular proliferation and the growth of reproductive tissues, estrogen takes part in the homeostasis of skin, the skeleton, the cardiovascular and central nervous systems, carbohydrate and lipid metabolism, and electrolyte balance [50].

The estrogen receptors (ERs), including both nuclear receptors (e.g., ERα and Erβ) and membrane-bound receptors (e.g., G protein-coupled estrogen receptors, GPER, also known as GPR30), are expressed in numerous innate and adaptive immune cells. The estrogen pathway includes genomic ligand-dependent signaling and nongenomic membrane-initiated signaling [51]. Classical nuclear receptors bind to lipophilic ligands, such as steroid hormones, forming ER dimers, which interact with specific DNA sequences, called estrogen response elements (EREs), located in the regulatory regions of target genes. On the other hand, membrane-bound receptors promote protein phosphorylation and protein kinase activation (e.g., mitogen-activated protein kinase, MAPK, and phosphatidylinositol 3-kinase, PI3K), impacting gene expression [52].

In both mice and humans, physiological levels of estradiol promote the pro-inflammatory activity of monocytes and macrophages, whereas higher levels—as observed in late pregnancy—are responsible for an anti-inflammatory shift [53]. In response to estrogen, macrophages help in building a tolerant immune environment for cell growth and tissue remodeling in the female reproductive tract [54]. Since they express receptors for sex hormones like ERs, human macrophages are subject to hormonal modulation [19]. For instance, estrogen contributes to macrophage M2 polarization via ERα [55]. Alternative (pro-healing) M2 macrophage activation is crucial for the wound healing process [56]. Similarly, estrogen treatment of murine bone marrow-derived macrophages induced augmented IL-4-induced M2 gene expression related to allergic lung inflammation. This can explain the observation that asthma progressively develops in women during adulthood, following the modification of the hormones [57]. Moreover, based on the pathway involved and the presence of intrinsic or extrinsic coregulators, estrogen alternatively enhances or suppresses the macrophage cytokine production (e.g., TNF, IL-1, and IL-6) [58]. As underlined by Straub et al., estrogen may exert both inflammatory or anti-inflammatory roles, depending on several criteria, including the immune stimulus, the involved cell types, the target organ, the hormone concentration, the timing of estrogen administration, and the variable expression of ERs [59]. Enright and Werstuck studied the influence of sex hormone supplementation on mouse bone marrow-derived monocytes differentiated into macrophages using macrophage colony-stimulating factor (M-CSF). The study revealed that 17β-estradiol inhibits inflammatory marker production in pro-inflammatory macrophages and enhances migration and the production of anti-inflammatory markers in anti-inflammatory macrophages, which resulted in a hypothesis that estradiol is protective in atherogenesis [60]. In a preclinical model of imiquimod-induced psoriasis, Adachi et al. detected more significant inflammation with higher concentrations of IL-1β and IL-17A in mice without endogenous ovarian hormones. Exogenous estradiol suppressed inflammation by inhibiting the production of IL-1β by neutrophils and inflammatory macrophages. In fact, this effect was not observed in mice not expressing ERs in macrophages and neutrophils [61].

Furthermore, estrogen modulates dendritic cell (DC) function through Toll-like receptor (TLR) signaling. Estradiol improves the production of pro-inflammatory cytokines, like type I IFN and TNFα, via TLR4 and TLR9 activation, and induces MHC-II and CD86 expression in conventional dendritic cells (cDCs). ERα deficiency significantly reduces type I IFN production and IRF5 expression, which are essential for TLR signaling in plasmacytoid dendritic cells (pDCs) [62]. It is worth noting that type I IFNs are essential in antiviral defense, because they ultimately activate the expression of interferon-stimulated genes (ISGs), which are involved in antiviral activities within infected cells [63]. In addition, as highlighted by Klein and colleagues, estrogen takes part in limiting the “cytokine storm” following influenza infection, clearing the virus without seriously damaging the host [64,65]. During the SARS-CoV-2 pandemic, estrogen has proven its ability to protect patients from death by inhibiting inflammation, suggesting the potential therapeutic role of medications boosting the downstream signaling of ER1/ER2 in COVID-19 [66]. There is also evidence of a consistent pro-inflammatory response to gut infections in men, frequently surpassing that in women. For example, typhoid ileal perforation is more common in men, a severe *Salmonella typhi* infection complication. Variations in hormonal control between sexes have been suggested, with estrogen potentially allowing the production of anti-inflammatory cytokines [67].

Estrogen also controls neutrophil apoptosis, chemotaxis, and the excretion of neutrophil extracellular traps (NETs), which participate in pathogen capture and cell death [53]. In humans, neutrophils spontaneously progress towards apoptosis both in vitro and in vivo. Molloy et al. observed delayed in vitro apoptosis in neutrophils derived from healthy women of reproductive age, as compared to men, as well as a further delay in cell death after the addition of physiological levels of estradiol and progesterone in both male and female neutrophils [68]. Dai et al. reported an increased number of neutrophils and neutrophil serine proteases (e.g., neutrophil elastase, proteinase 3, and cathepsin G) in the spleens of estrogen-treated mice, as well as in autoimmune-prone female mice [69]. During the time of ovulation in women, neutrophils overexpress ERα and ERβ, and estrogen stimulates them to increase the expression of neuronal nitric oxide synthase (nNOS). Interestingly, Molero et al. reported that in vitro treatment of men’s neutrophils with 17β-estradiol also increases the expression of ERα and nNOS proteins [70]. Neutrophils from men of reproductive age are less energetic than women’s neutrophils, with the differences being hormone-dependent. RNA sequencing showed female neutrophils to have a greater number of type I IFN-stimulated genes that enhance the TLR agonist response, while neutrophils from men were elevated in mitochondrial metabolism [71].

ERα and ERβ are present in NK cells, and high levels of estradiol inhibit NK cell activity in wild-type mice and ERα-deficient mice [72]. Hao et al., using a mouse model, reported that estradiol increases the count of NK cells, possibly by increasing the expression of mini-chromosome maintenance proteins MCM7 and MCM10. It suppresses NK cell cytotoxicity by downregulating activating receptors and reducing the expression of granzyme B and FasL [73]. Similarly, decreased NK cell cytotoxicity was observed in postmenopausal women receiving high doses of estradiol [74]. However, enhanced cytotoxic activity of human NK cells treated with estradiol against tumor cells emerges in certain in vitro studies. This effect is antagonized by tamoxifen, a selective estrogen receptor modulator [75].

ERs are expressed in T and B lymphocytes, with various expression levels according to the cell type. For example, CD4^+^ T cells have higher levels of ERα, B cells express more ERβ, and mature lymphocytes express both receptors. Estrogen takes part in the survival and proliferation of B lymphocytes, antibody production (including the switch to IgE involved in allergic inflammation), and the expression of molecules regulating apoptosis and autoimmunity [76].

High levels of estrogen may determine thymic involution, due to the reduction in early thymic progenitors and the inhibition of thymocyte proliferation [77]. Estrogen promotes T helper cell (Th) 1 differentiation, IFN-γ, and IFN-γ-dependent inflammation [78]. As observed by Maret and colleagues, administering estrogen to ovariectomized mice increases antigen-specific CD4^+^ T cell responses and the development of IFN-γ-producing cells. The researchers noticed the essential role of ERα expression in hematopoietic cells for Th1 responses [79]. Ovariectomized female mice treated with estradiol exhibited improved immune responses against malaria infection, consisting in increased levels of IFN-γ and IL-10 during peak parasitemia, and increased IgG1 responses, together with a mild clinical presentation [80]. Khan et al. found that estrogen upregulated the levels of IL-17, a pro-inflammatory cytokine involved in several inflammatory and autoimmune diseases, and its specific transcription factor retinoic acid-related orphan receptor (ROR)γt in activated mouse splenocytes, whereas IL-27 and IFN-γ play a suppressive role [81].

Estrogen plays a key role in the development and proliferation of T regulatory (Treg) cells [82], contributing to immune tolerance at the maternal–fetal interface. As has emerged from mouse studies, both in vivo and in vitro, estradiol at physiological doses can increase the number of Tregs in different tissues, as well as the expression of Foxp3 and IL-10 genes, thus suggesting its role as a regulatory factor during the implantation period [83]. Starting with the evidence that sex differences in pneumonia survival exist, with a higher mortality in males, Xiong et al. evaluated the impact of estradiol on pulmonary damage resolution in a mouse model of pneumococcal pneumonia. The authors highlighted that estradiol promotes the expansion of Tregs and Treg expression of Foxp3, CD25, and GATA3 via ERβ signaling, playing a crucial role in recovery from lung injury [84]. Treatment with estradiol of Calu-3 respiratory epithelial cells increased IgA transcellular transport and the expression of TLR-4. Secretory IgA regulated by TLR-4 is required in respiratory secretions, giving protection against the transit of bacteria and potentially preventing pneumonia [85].

Estrogen enhances antibody and autoantibody responses, with women generally exhibiting higher antibody levels following infection or vaccination [86]. As shown by Mai et al., estrogen–ER complexes bind to EREs in the HOXC4/HoxC4 promoter, inducing the expression of HoxC4 transcription factor, which upregulates AID (activation-induced cytosine deaminase), involved in antibody class switch DNA recombination and somatic hypermutation [87]. In mouse B cells, a proliferation-inducing ligand (APRIL) increases the expression of both HoxC4 and AID, whereas B cell-activating factor, belonging to the TNF family (BAFF), increases the expression of AID only. In this context, NF-κB enhances the APRIL-dependent transcription of HoxC4 [88].

### 3.2. Progesterone

Progesterone, a sex steroid hormone, is essential for initiating and maintaining pregnancy, which requires maternal adaptations, such as tolerogenic immunological changes in the context of the blood–placenta barrier [89]. The progesterone receptor (PR) belongs to the family of nuclear receptors and acts as a ligand-dependent transcription factor [90]. The activation of PR, following its binding with the related steroid hormone, modulates the expression of target genes. Among the multiple isoforms of PR that have been described, the best-characterized ones are PR-A and PR-B, regulating different genes [91]. PRs can be found in most immune cells, such as epithelial cells, macrophages, dendritic cells, and lymphocytes, and progesterone can also bind to glucocorticoid and mineralocorticoid receptors [92].

Sex steroid hormones, particularly estradiol and progesterone, influence both innate and adaptive immunity against sexually transmitted infections in the human female reproductive tract [93]. These hormones are responsible for changes in the cytokine levels in cervico-vaginal fluids occurring during the menstrual cycle. For instance, the depletion of progesterone leads to the upregulation of IL-8, MCP-1 (monocyte chemoattractant protein-1), and COX-2 (cyclooxygenase-2) in the human endometrium [94]. It has been observed that progesterone induces the expression of ISGs in the epithelium of the ovine uterus [95]. Gómez-Oro et al. found that progesterone promotes macrophage–neutrophil crosstalk in the human female reproductive tract, thus potentiating chemokine (C-X-C motif) ligand 2 (CXCl2)-dependent neutrophils’ killing of sexually transmitted and opportunistic microorganisms [96]. Progesterone activates innate immune pathways in ECC-1 cells (human endometrioid endometrial cancer cell line) derived from the human uterine endometrium, thus increasing resistance to Chlamydia infection, compared to estradiol-treated cells. An upregulation of genes encoding, among others, CC and CXC chemokines, IL-17C, TNF-α, and type I and II interferon receptors was reported in progesterone-treated and *Chlamydia trachomatis*-infected cells [97]. Li et al. examined progesterone treatment impacts on immune response and cell death in Calu-3 (lung adenoma) and ACH-3P (extravillous trophoblast) cells infected by influenza A virus. Progesterone increased cytotoxicity and inflammatory cytokine expression in Calu-3 cells but reduced cytotoxicity in ACH-3P cells. Progesterone also reduced caspase-1 activity in both cell lines, suggesting that it has an inflammasome-independent protective effect in viral infection in the placenta [98]. Groh and colleagues reported the ability of progesterone to inhibit TNFα and IL-6 production within trained innate immunity following the exposure of monocytes to oxidized low-density lipoproteins (oxLDLs), thus contributing to cardiovascular protection in premenopausal women [99]. However, Earhart et al. described reduced trained immunity in female mice, which depended on progesterone and was rescued by progesterone receptor antagonism, and which was responsible for sex bias in survival from opportunistic infections (e.g., bacteremia of *Burkholderia gladioli*) [100].

Metabolomic analysis conducted on both control and virus-infected mice revealed that serum progesterone levels significantly increase after viral infections via the hypothalamic–pituitary–adrenal axis. Progesterone promotes innate antiviral response through PGR-dependent activation of the tyrosine kinase SRC, which mediates phosphorylation of transcriptional factor IRF3 that induces, when activated, antiviral genes, such as IFNB1 [101]. By analyzing hormone testing reports of SARS-CoV-2-infected men compared to healthy ones, Su et al. found that SARS-CoV-2 infection increases progesterone levels, which are positively associated with lower severity of COVID-19 [101], thus suggesting the immunomodulatory role of progesterone in infectious diseases. Hall et al. treated ovariectomized adult female mice with exogenous progesterone, which enhanced protection against influenza A virus infection, modifying the inflammatory environment of the lungs. Progesterone increased IL-6, TGF-β, and regulatory Th17 cells and upregulated the epidermal growth factor amphiregulin (AREG), facilitating lung recovery from the infection [102]. In addition, influenza A virus infection has been associated with lower progesterone levels in pregnant mice. In this context, progesterone therapy may be a useful tool for alleviating the effects of viral infection, as well as preventing pregnancy complications [103]. Nevertheless, progesterone can inhibit inflammatory responses against bacterial infections (e.g., *Escherichia coli*) in bovine endometrium. Bovine endometrial epithelial cells (BEECs) cultured with progesterone, lipopolysaccharide (LPS), or *Escherichia coli* alone showed increased expression of inflammatory cytokines. Progesterone, however, inhibited LPS- or *Escherichia coli*-induced cytokine expression by blocking NF-κB activation and MAPK phosphorylation in BEECs [104].

Progesterone is involved in the differentiation of fetal T cells into different T cell types in cord blood (CB). As noticed by Lee et al., progesterone promotes CB T cell differentiation into Tregs expressing FoxP3, an effect not observed in peripheral blood (PB) adult T cells. Moreover, in progesterone-rich CB, the differentiation of CD4^+^ T cells into inflammation-associated Th17 cells is suppressed, since progesterone increases IL-2-dependent STAT5 (signal transducer and activator of transcription 5) activation and since it inhibits IL-6-dependent STAT3 activation as well as the expression of the IL-6 receptor by T cells [105]. In general, progesterone promotes Th2-type responses in CD4^+^ T helper cells rather than Th1-type responses, leading to the release of anti-inflammatory cytokines such as IL-4 and IL-10 [92]. At the maternal–fetal interface, progesterone impacts the cytotoxic activity of decidual lymphocytes, blocking the expression of perforin [106]. As observed in rhesus macaques, the concentrations of IgG and IgA in cervico-vaginal lavage vary according to the stage of the menstrual cycle, with higher levels during the menstrual phase and early follicular phase and lower levels around ovulation, in response to hormonal changes [107]. However, the same researchers found a higher frequency of immunoglobulin- and antibody-secreting cells (ISCs and AbSCs) in cervico-vaginal tissues and lymphoid tissues of primates in the periovulatory stage [108]. In vitro studies on rhesus monkey peripheral blood mononuclear cells (PBMCs) showed that progesterone has an inhibitory effect, whereas estrogen has a stimulatory effect, on immunoglobulin-secreting cell (ISC) frequency. Importantly, B cell immunity in female rhesus macaques and, by extension, women depends on the presence of CD8^+^ T cells regulating Ig secretion in response to sex hormone levels [108].

### 3.3. Androgens

Androgens and the androgen receptor (AR) regulate the development and functions of reproductive systems in both sexes. The AR, encoded by a gene located on the X chromosome, is a nuclear receptor encompassing four different domains: the N-terminal domain, ligand-binding domain, DNA-binding domain, and C-terminal domain. After binding testosterone or 5α-dihydrotestosterone (DHT), the translocation of AR into the nucleus occurs, and its subsequent interaction with androgen responsive elements (AREs) nearby target genes impacts their expression [109]. Androgens can also bind membrane ARs, which are G protein-coupled receptors activating intracellular signaling cascades, such as mitogen-activated protein (MAP) kinases [110].

ARs are present in numerous innate and adaptive immune cells, such as macrophages, monocytes, neutrophils, B cells, and T cells. As a result, androgens exert a direct effect on immunity by regulating the expression of immunoregulatory genes through both DNA-binding-dependent and -independent mechanisms [48]. Several studies conducted on animals have clarified the impact of androgens on macrophage and monocyte function. In vitro, the treatment of mouse-derived macrophage cell lines with testosterone induced the production of IL-10, augmented by concomitant lipopolysaccharide (LPS) stimulation, together with a reduction in nitric oxide (NO) and TNF-α, developing anti-inflammatory macrophages [111]. Similarly, testosterone decreased TLR4 protein expression, reducing the response to inflammatory stimuli [112]. On the contrary, increased TLR4-expressing macrophages were identified in orchiectomized mice, which more likely developed endotoxic shock, resulting from a generalized inflammatory response induced by systemic infection with Gram-negative bacteria [112].

Thompson et al., exposing murine bone marrow-derived DCs to LPS and DHT, identified a dose-dependent downregulation of IL-6 and upregulation of TNF, IL-4, and IL-10, as expected from DCs with an anti-inflammatory phenotype [113]. Sex disparities in pDC response to HIV-1 reveal that female pDCs produce more IFN-α when TLR7 is activated, more robustly stimulating CD8^+^ T cells. This could account for reduced viral loads and more rapid disease progression in women following early HIV-1 infection [114]. Using a mouse model, female DCs were found to be more activated, whereas orchidectomy of male mice increased antigen-presenting cell activation following infection [115]. Chuang et al. reported that AR-knockout male mice were neutropenic and more vulnerable to bacterial infections, thus hypothesizing the role of androgens in the development and activity of neutrophils, probably by enhancing granulocyte colony-stimulating factor signaling [116]. However, Scalerandi et al. noticed in testosterone-induced neutrophils weakened bactericidal ability and decreased myeloperoxidase activity, whereas the production of anti-inflammatory cytokines (e.g., IL-10 and TGFβ1) was upregulated [117]. Hreha et al. recognized a higher severity of urinary tract infections attributable to androgens. Female mouse studies highlighted that androgen exposure hindered the maturation of neutrophils within the urinary tract, with reduced ability in both degranulation and phagocytosis, thus elevating the likelihood of ongoing infections and the formation of abscesses [118]. This skewed activation of neutrophils by androgen may contribute to explain why males have higher risk for sepsis and lower ability to fight infections, as compared to females [110,119].

Androgens suppress the development, tissue maintenance, and function of lung innate lymphoid cells (ILCs) and promote an anti-inflammatory phenotype with increased ST2 (suppression of tumorigenicity 2) and KLRG1 (killer cell lectin-like receptor G1) and decreased CD25. This may help to explain the poorly understood lower airway inflammation observed in male mice and, more generally, the lower prevalence of asthma in males compared to females [120]. Moreover, ILCs are involved in the regulation of type 2 inflammation in response to parasitic infections [121].

In the literature, several studies describe the role of androgen/AR in suppressing thymic development, as demonstrated by thymic enlargement that characterizes AR knockout (ARKO) and castrated mice [122,123], with an increased release of naïve CD4^+^ and CD8^+^ T lymphocytes into the bloodstream. Androgens exert a negative effect on T cell proliferation, apparently due to the inhibition of IL-2 signaling in T cells. In vitro, a more vigorous proliferation of T cells derived from castrated mice was observed, as compared to T cells from normal mice [124]. The inhibitory effect of androgens on T cell proliferation may contribute to sex differences in autoimmune responses. Kissick et al. investigated the effects of androgen deprivation on T lymphocytes from the spleen of castrated male mice in vivo and in vitro. Significant gene expression changes in pathways involved in IFN signaling and Th1 differentiation were identified, revealing that testosterone inhibits IL-12-induced STAT4 phosphorylation [125]. As noticed by Fijak et al., the exposure of splenic CD4^+^ T lymphocytes to androgens leads to an upregulation of Foxp-3 and IL-10, an effect that can be reversed by the addition of flutamide, an anti-androgen [126].

It has been observed that AR is expressed in B cell progenitors rather than in mature B cells, implying the influence of androgens on B cell development [127]. The absence of AR in B cells has been associated with B cell expansion in blood and bone marrow, resulting from enhanced proliferation and resistance to apoptosis. In addition, AR-deficient mice in B lymphocytes also had elevated levels of serum IgG2a and IgG3 [128]. Furthermore, testosterone influences BAFF, a crucial cytokine for the sustained existence of B cells. Wilhelmson et al. demonstrated that a lack of testosterone leads to increased BAFF levels in serum, as well as higher counts of BAFF-producing fibroblastic reticular cells in the spleen. Similarly, men with low testosterone levels also show heightened serum BAFF levels. Taken together, these data suggested the involvement of BAFF in testosterone-dependent protection against autoimmunity in men [129].

In the last few years, the role of androgens in the pathogenesis of SARS-CoV-2 infection has been investigated, since a negative course of the disease is seen more frequently in men worldwide, whereas preadolescent children are generally asymptomatic. Androgens can influence the penetration of SARS-CoV-2 into host cells, by promoting the expression of transmembrane protease serine 2 (TMPRSS2) [130]. Nevertheless, low serum levels of testosterone seem to be responsible for poor prognosis and death in elderly men with COVID-19, confirming the double-edged sword role of sex hormones in metabolic equilibrium and immune regulation [130,131].

The major effects of sex steroid hormones on the immune system are listed in Table 2.

## 4. Sex-Related Differences in the Immune Response to Infections During Aging

Immunosenescence, gradual changes in the immune system that occur with age, is complex, featuring a decline in some aspects of immune function, increasing stable or even hyperactivated elements, and considerable variability between individuals. A broader term for this phenomenon is “immune aging”. These features are characterized by decreased responsiveness to new antigens and vaccines, caused by reduced naïve T and B lymphocyte numbers and increased accumulation of memory cells [132]. Older adults are persons with different immunology driven by a life-long collection of heterogenic contacts to environmental, infectious, and lifestyle factors, known as immunobiography [133]. Immune aging is influenced by factors across the life course, described in immunobiography, including the phenomenon known as trained immunity, which provides potential mechanisms that may aid in accounting for the substantial heterogeneity observed in immune parameters, particularly among centenarians [134]. Therefore, multiple biological and environmental modulators, such as the mode of delivery, early life nutrition, the microbiome profile, and socioeconomic factors, including dietary patterns, occupation-related exposures, and educational attainment, exert their cumulative impact on the development and function of both innate and adaptive immune cells across an organism’s lifetime. These factors affect the immune system’s ability to respond to internal and external aggression, with the impact being felt especially in old age [135].

Aging brings about notable transformations in both innate and adaptive immune cells. One cornerstone of immune aging, also known as “inflammaging”, is the chronic activation of innate immune cells and persistent inflammatory responses that last for years [136]. Inflammaging is associated with a deregulated systemic cytokine profile, including an elevated concentration of important pro-inflammatory cytokines (IL-1, IL-6, and TNF-α) and decreased expression of the anti-inflammatory cytokines (IL-10 and TGF-β). This imbalance of the inflammatory milieu contributes to an increased susceptibility to infections and the progression of chronic illnesses, thereby leading to increased morbidity and mortality in the elderly [137].

Research in mice indicates that aging dramatically impacts macrophage function, one of the critical components of the innate immune response. Macrophages from aged mice secrete less pro-IL-1β than their younger counterparts, leading to impaired inflammatory regulation and reduced antimicrobial functionality [138]. This reduction in cytokine generation reflects a general lowering in immune efficacy as one ages. Aged mice exhibit a change in the expression of the IFN-γ responsive gene [139]. IFN-γ is a crucial cytokine involved in immunosurveillance and tissue repair. Also, there are structural changes in neutrophils in older people, such as decreased plasma membrane fluidity and a reduced ability of neutrophils to adhere to the vascular wall, also contributing to decreased function of neutrophils as a whole [140]. Interestingly, although senescent neutrophils are generally more phagocytic than neutrophils from younger individuals, increased phagocytic activity does not always correlate with improved immunity [141].

Neutrophils are directed to inflammatory sites via the chemokine receptor CXCR2, which leads to a prompt and efficient immune response. Yet, Weisel et al. showcased that neutrophils from older subjects have markedly lower CXCR2 expression. This age-related decline could hamper neutrophil migration to inflamed tissues, thus leading to impaired inflammatory responses in the elderly, which impairs neutrophil migration [142]. Such reduced expression impairs cellular response to chemotactic signals, which dampens the ability of these cells to migrate to and perform their action at sites of infection or tissue damage, thus blunting the inflammatory response. In addition, aging disarms the trafficking properties of neutrophils and renders them less effective at migrating and invading sites of infection, which diminishes the effectiveness of immune surveillance and response [143]. Impaired clearance of *Escherichia coli* in septic peritonitis was revealed in aged mice who, nonetheless, exhibited increased recruitment of neutrophils in peritonitis caused by LPS but not aseptic peritonitis [144]. Importantly, even where neutrophils were recruited, they failed to function appropriately, and aged mice had a significantly reduced capacity to kill *E. coli* as well as markedly reduced generation of reactive oxygen species (ROS) in response to LPS stimulation [145].

DCs function is also impaired during aging, with decreased antigen presentation and cytokine production [146]. Aging macrophages have reduced counts and functionality of myeloid and plasmacytoid DC as well as impaired activation of T cells and poor antiviral activity [147].

Regarding adaptive immunity, the B cell compartment undergoes major changes, with age-associated B cells (ABCs) proliferating with age and driving autoimmune activity and inflammaging. Aging induces immune sexual dimorphism via excess IL-7 signaling and N-glycan branching, leading to diminished T cell activity of older women [148]. ABCs, prompted by TLR activation alongside co-stimulatory signals, modulate the immune responses to pathophysiologic outcomes associated with autoimmune diseases and chronic inflammatory processes in the elderly [149]. Moreover, NK cell cytotoxicity and cytokine production are dampened during aging, despite the accumulation of long-lived NK cells in older adults. Such functional decline is associated with greater vulnerability to infections and cancers. Altered cytokine release by aged NK cells may also help drive more systemic defects in immunity, including a blunted adaptive immune response and accumulating senescent cells [150].

Recent studies have demonstrated that men have a significantly accelerated immune aging process compared to clinically matched women, leading to the worsening of both innate and adaptive immunity with age. For example, sex hormones may play a significant role in explaining the above disparity in immunosenescence between men and women. Sex steroids use different strategies during the reproductive years to regulate the immune response: estrogen upregulates immune activity, while progesterone and androgens downregulate it. Estrogen receptors are not just limited to reproductive tissues; however, they are also on many of the immune cells, including lymphocytes, monocytes, macrophages, and dendritic cells. This observation could be explained by the fact that the drop in estrogen following menopause can strongly influence the functions and regulation of immune cells [151,152]. Consequently, postmenopausal women observe an advanced rate of immunosenescence, thus reversing the possible immunological advantage women physiologically experience during their reproductive years. This change may help explain why the immune systems of older women may begin to track, or even trail, those of equally aged men [153].

In one study by Márquez et al., PBMCs from 172 healthy volunteers aged between 22 and 93 years were examined via ATAC-seq, RNA-seq, and flow cytometry. The research revealed a common epigenomic aging signature, which was marked by a reduction in naïve T cells and enhanced activity of monocytes and cytotoxic cell types. The age-related changes were stronger in men and were associated with a loss of B cell-specific genomic regions that was male-specific [148].

Aging and immune response research identifies that older individuals, particularly men, have compromised responses to vaccines like the influenza vaccine, which has weaker effects in this population. As the world’s population ages, immunosenescence is becoming a significant public health issue. The research identifies that biological sex is significant in immune system aging [136].

Unfortunately, there is no transcriptomic profiling of old, very aged, or young immune cells that show sex differences in immune aging. Some studies show that, different from men, a considerable number of pathways were modified in older women, which included decreased T cell-mediated defense and inflammatory signatures [154]. In support of this, Jentsch-Ullrich et al. documented higher CD4^+^ T cell and lower NK-cell counts in elderly women compared to men [155]. However, reference ranges for these immune cells in postmenopausal women tend to converge with men. More evidence of sex-based differences can be found in a publication by Hirokawa et al. indicating that immunosenescence advances faster in men [156]. This was mirrored by a sharper drop in total T cells, particularly naïve CD4^+^ and CD8^+^CD28^+^ subsets, as well as B cells, T cell proliferative capacity, and IL-6 production. In contrast, men exhibited a muted age-associated expansion of memory CD4^+^ T cells and NK cells in comparison with women. Sex also seems to affect cytokine secretion during aging. For instance, older men had reduced production of IFN-γ and IL-17 after in vitro stimulation, which was preserved in older women [157]. Conversely, one study found increased IL-10 secretion in older women, but not in older men, suggesting a complex, sex-specific remodeling of immune function with age [158].

## 5. Conclusions and Future Prospectives

As highlighted in our review, sex differences play a crucial role in the immune response to infectious diseases, yet they remain insufficiently explored. Importantly, effective immunity relies on maintaining balance. It must remove pathogens without hurting the body. Disruption of this equilibrium generates an overactive or underactive immune response. This reaction could present with autoimmune disease or immune deficiencies. Understanding sex-based differences in the immune system can enhance our knowledge of various diseases [159] and contribute to the development of more personalized and effective medical treatments. As research continues regarding where immune mechanisms diverge based on sex, we will have a clearer picture that will help us implement valid, gender-specific clinical strategies.

LOY is a new genetic topic. Recent studies focus on the Y chromosome and its role in health problems. They often note the loss of the chromosome or gene silencing from hypermethylation. However, researchers seldom study the Y chromosome looking at the whole genome. Because a background of interacting genetic mutations influences what happens with disease, as do age and sex, just examining the Y chromosome alone is not enough [160]. The Y chromosome and its variants may change how genes on autosomes are expressed. Research on fruit flies indicates the Y chromosome can affect autosomal gene expression by changing heterochromatin patterns throughout the genome [161]. Several studies have shown a link between mosaic LOY and various diseases in men, including Alzheimer’s disease, kidney disorders, cardiovascular conditions, and different forms of cancer [162]. It would be valuable to conduct further research to determine whether this phenomenon also affects the immune response to infectious diseases.

In general, women detect various pathogens and recruit innate immune cells better than men, activating the adaptive immune responses earlier than men. Estradiol at low levels favors Th1 responses and cell-mediated immunity, while estradiol at higher levels favors Th2 responses and humoral immunity. Progesterone and testosterone equally modulate a broad range of anti-inflammatory effects, and they both suppress innate immune activity. The woman’s hyperimmune response also depends on X-linked genes, which may moderate the severity of infection. Men are generally more susceptible to some bacterial and parasitic infections than women, although susceptibility varies from pathogen to pathogen [163].

However, the mere explanation of infectious disease mechanisms is limited when viewed solely through the lens of biological sex differences; such considerations must also extend to the gender side, including social and behavior-related aspects.

It is imperative that modern science acknowledges the intricacy and heterogeneity of sex-based biology, particularly in the context of transgender individuals undergoing gender-affirming hormone therapy (GAHT), as well as those with genetic syndromes or disorders of sexual development [45]. For instance, Lakshmikanth et al. have recently published their research evaluating the immunological consequences of gender-affirming testosterone therapy in trans men [164]. Testosterone augments monocyte responses by raising TNF, IL-6, and IL-15 levels, increasing NF-κB-regulated gene activation, and increasing IFN-γ responses in NK cells. Investigating the effect of GAHT on the immune system may also provide data on the sex-dimorphic immune responses in cisgender individuals [164].

Finally, unlike what one would commonly expect, sex differences in immune response persist with age. Recent findings suggest that while men’s mortality risk steadily increases with age, those over 90 tend to become more resilient compared to women of the same age [165]. On the other hand, some studies found no major differences in in-hospital mortality rates among male and female geriatric patients [166]. The phenomenon of inflammaging is influenced heavily by hormonal action, genetic control, epigenetic differences, and social challenges. During their reproductive years, women have a much stronger immune function; however, this advantage is lost postmenopause due to increased levels of immunosenescence as estrogen levels drop. While women generally live longer than men, they often experience poorer health in later life [167]. The close relationship between aging and reproduction suggests that both biological and socio-environmental factors contribute to sex differences in health span, potentially aiding in the identification of sex-specific biomarkers of aging.

## Figures and Tables

**Figure 1 diseases-13-00179-f001:**
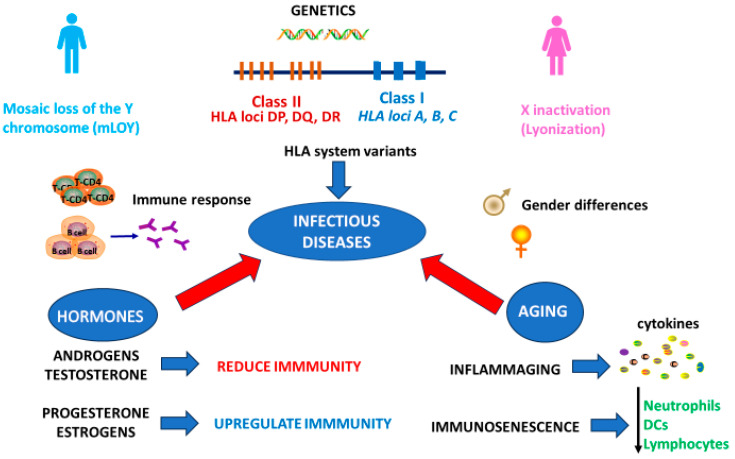
Sex-related biological (genes, hormones, immune response) factors modulate the risk, severity, and outcomes of infectious diseases. Genetic inheritance (XX vs. XY) and gene status can either increase or reduce innate and adaptive immune responses. In addition, hormonal differences (estrogen, progesterone, androgens) can also enhance or decrease the activity of macrophages, neutrophils, NK, and T and B lymphocytes. The complex biological mechanism interacts with social gender, demonstrating the need for a sex- and gender-based precision medicine initiative.

**Table 1 diseases-13-00179-t001:** Major genes and pathways involved in immune regulation that are regulated by the corresponding sex chromosomes.

Chromosome	Regulated Genes and Pathway	Function	Reference
*Y*	IMD	Regulates innate immune response via NF-κB signaling (in Drosophila model)	[8]
*Y*	Expression of CD99	Facilitates immune cell transendothelial migration and adhesion	[14]
*X*	FOXP3	Master regulator of T regulatory (Treg) cell development and function	[16]
*X*	CD99, CXCR3	CD99: immune cell migration; CXCR3: chemokine receptor involved in leukocyte trafficking	[16]
*X*	IL-2RG	Common gamma chain in several interleukin receptors, essential for T and NK cell development	[16]
*X*	OGT, CYBB	OGT: protein glycosylation in signaling; CYBB: component of NADPH oxidase complex for pathogen killing	[16]
*X*	TLR7	Recognizes single-stranded RNA viruses, activates innate immune signaling	[17]
*X*	IRAK1	Mediates downstream TLR/IL-1 receptor signaling, promotes inflammatory cytokine production	[20]
*X*	BTK	Regulates B cell development, activation, and NF-κB signaling	[22]
*X*	XEDAR	TNF receptor family member, activates both canonical and non-canonical NF-κB pathways	[23]

Abbreviations: IMD, immune deficiency; NF-κB, nuclear factor kappa B; CD99, cluster of differentiation 99; FOXP3, forkhead box protein 3 gene; CXCR3, C-X-C motif chemokine receptor 3; IL-2RG, interleukin-2 receptor subunit gamma; OGT, O-linked N-acetylglucosamine transferase; CYBB, cytochrome B-245 beta chain; NADPH, nicotinamide adenine dinucleotide phosphate; TLR7, Toll-like receptor 7; IRAK1, interleukin-1 receptor-associated kinase 1; BTK, Bruton tyrosine kinase; XEDAR, X-linked ectodermal dysplasia receptor; TNF, tumor necrosis factor.

**Table 2 diseases-13-00179-t002:** Major changes in immune cells and pathways induced by sex steroid hormones.

Sex Hormones	Effects on Immune Cells and Pathways	References
Estrogen	Macrophage M2 polarization	[55]
	Modulation of cytokine production and limiting “cytokine storm”	[58,64,65]
	Promotion of type I IFNs and TNFα through TLR signaling	[62]
	Neutrophil apoptosis, chemotaxis, and release of NETs	[53]
	Increasing nNOS	[70]
	Increasing the number of NK cells and suppressing their cytotoxicity	[73,74]
	Survival and proliferation of B lymphocytes	[76]
	Antibody and autoantibody production	[76]
	Th1 differentiation	[78]
	IFN-γ-dependent inflammation	[78]
	Differentiation and expansion of Tregs	[82]
Progesterone	Promotion of macrophage–neutrophil crosstalk	[96]
	Inhibition of TNFα	[99]
	Expression of antiviral genes (e.g., IFNB1)	[101]
	T cell differentiation into Tregs	[105]
	Th2-type responses	[92]
	Release of anti-inflammatory cytokines (e.g., IL-4 and IL-10)	[92]
	Inhibition of ISCs frequency	[108]
Androgens	Induction of IL-10 and reduction in NO and TNFα	[111]
	Induction of anti-inflammatory macrophages and dendritic cells	[111,113]
	Reduction in TLR4 expression	[112]
	Neutrophil recruitment with weakened bactericidal ability	[116,117]
	Inhibition of ILCs	[120]
	Suppression of thymic development	[122,123]
	Inhibition of T cell proliferation	[124]
	Inhibition of IL-12-induced STAT4 phosphorylation	[125]
	Negative control on BAFF	[129]

Abbreviations: IFN, interferon; TNFα, tumor necrosis factor α; TLR, Toll-like receptor; NETs, neutrophil extracellular traps; nNOS, neuronal nitric oxide synthase; NK, natural killer; Th, T helper; IFNB1, interferon beta 1; Tregs, T regulatory cells; IL, interleukin; ISCs, immunoglobulin-secreting cells; ILCs, innate lymphoid cells; STAT, signal transducer and activator of transcription; BAFF, B cell-activating factor belonging to the TNF family.

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
