# Peer review of "Sex Differences in Immune Responses to Infectious Diseases: The Role of Genetics, Hormones, and Aging"

_diseases, 2025, doi:10.3390/diseases13060179_

Round 1
Reviewer 1 Report
Comments and Suggestions for Authors
The MS is well written and organised, but the manuscript requires some modification. few are
- what is " sex steroid hormones" in abstract?
- In subheading 2. There are many 2-3 line paragraphs. Please merge these paragraphs. Same comment for the rest of the subheadings
- In Table 1. Add a column stating the function of regulatory genes
Accept after minor modifications
Author Response
Rome, June 04, 2025
Dear Editor of Diseases
First, my coauthors and I would like to thank you sincerely for this opportunity to cooperate. We profoundly thank the reviewers for the comments and useful suggestions to improve the paper. We thank You for your constructive critique and hope the review process has improved the manuscript. If additional changes are warranted, we will make them. 
We hope that this revised version of our manuscript may now be found suitable for publication. 
This is a point-by-point list of changes made in the paper:
Reviewer 1
The MS is well written and organised, but the manuscript requires some modification. few are
- What is "sex steroid hormones" in abstract?
We specified the type of hormones, as requested. - In subheading 2. There are many 2-3 line paragraphs. Please merge these paragraphs. Same comment for the rest of the subheadings.
We have merged the 2-3 line paragraphs throughout the text, as requested. - In Table 1. Add a column stating the function of regulatory genes.
We added a column on regulatory genes, as requested.
We thank You for your constructive critique and we hope the review process has led to an improved manuscript.
If additional changes are warranted, we will make them.
We hope that this revised version of our manuscript may now be found suitable for publication.
Sincerely,
Rossella Cianci
Reviewer 2 Report
Comments and Suggestions for Authors
Dear authors,
The current review is well written and it affects a problem that has practical significance. The article could be published in its present version.
Author Response
Rome, June 04, 2025
Dear Editor of Diseases
First, my coauthors and I would like to thank you sincerely for this opportunity to cooperate. We profoundly thank the reviewers for the comments and useful suggestions to improve the paper. We thank You for your constructive critique and hope the review process has improved the manuscript. If additional changes are warranted, we will make them. 
We hope that this revised version of our manuscript may now be found suitable for publication. 
This is a point-by-point list of changes made in the paper:
Reviewer 2
The current review is well written and it affects a problem that has practical significance. The article could be published in its present version.
Thank you for the good comments.
We thank You for your constructive critique and we hope the review process has led to an improved manuscript.
If additional changes are warranted, we will make them.
We hope that this revised version of our manuscript may now be found suitable for publication.
Sincerely,
Rossella Cianci
Reviewer 3 Report
Comments and Suggestions for Authors
I found the review very complete and well done.
Just few points:
Chapt. 2 : ref 8 (Myllymäki et al.) showed that the Immune Deficiency (IMD) pathway, in Drosophila, is affected by Y-linked regulatory variation. Can you explain to me better in which way she showed this?
Chapt. 2 : by impairing the function of impaired immune cells
AVOID THE REPETITION
End of Chapt. 2 : you miss a full stop in the last but one period. since you had put the ref. and the full stop in the new paragraph. It should be better to begin the new paragraph with sentence beginning with In addition to sex chromosomes
Author Response
Rome, June 04, 2025
Dear Editor of Diseases
First, my coauthors and I would like to thank you sincerely for this opportunity to cooperate. We profoundly thank the reviewers for the comments and useful suggestions to improve the paper. We thank You for your constructive critique and hope the review process has improved the manuscript. If additional changes are warranted, we will make them. 
We hope that this revised version of our manuscript may now be found suitable for publication. 
This is a point-by-point list of changes made in the paper:
Reviewer 3
I found the review very complete and well done.
Just few points:
Chapt. 2: ref 8 (Myllymäki et al.) showed that the Immune Deficiency (IMD) pathway, in Drosophila, is affected by Y-linked regulatory variation. Can you explain to me better in which way she showed this?
We have made the concept explicit, as required.
Chapt. 2 : by impairing the function of impaired immune cells AVOID THE REPETITION.
We eliminated the repetition.
End of Chapt. 2 : you miss a full stop in the last but one period. since you had put the ref. and the full stop in the new paragraph. It should be better to begin the new paragraph with sentence beginning with In addition to sex chromosomes.
We have corrected the mistakes, as requested.
We thank You for your constructive critique and we hope the review process has led to an improved manuscript.
If additional changes are warranted, we will make them.
We hope that this revised version of our manuscript may now be found suitable for publication.
Sincerely,
Rossella Cianci